# Reporting adverse events related to medical devices: A single center experience from a tertiary academic hospital

Fahad Alsohime[1,2☯¤]*, Mohamad-Hani Temsah[1,2,3☯], Gamal Hasan[2,4], Ayman Al-Eyadhy[1,2], Sanaa Gulman[2], Haytam Issa[2], Omar Alsohime[5]

1 Pediatric Department, College of Medicine, King Saud University, Riyadh, Saudi Arabia, 2 Pediatric Intensive Care Unit, King Saud University Medical City, Riyadh, Saudi Arabia, 3 Prince Abdullah Ben Khalid Celiac Disease Research Chair, King Saud University, Riyadh, Saudi Arabia, 4 Department of Pediatrics, Assiut Faculty of Medicine, Assiut University, Egypt, 5 Standards and Guidelines Department, Medical Devices Sector, Saudi Food and Drug Authority, Riyadh, Saudi Arabia

☯ These authors contributed equally to this work.
¤ Current address: College of Medicine, King Saud University, Riyadh, Saudi Arabia
* falsohime@ksu.edu.sa

**Data Availability Statement:** All relevant data are within the paper and its Supporting Information files.

## Abstract

Intensive care units (ICU) rely on multiple technical resources with extensive use of different medical devices, such as ventilators, vital sign monitors, infusion, and injection pumps. This study explored how ICU nurses approach adverse events related to medical devices in a single tertiary center and identify their level of awareness of the national reporting system for adverse events related to medical devices beside their source for risk information updates. Totally, 297 nurses working in the ICU at King Saud University Medical City completed a survey on medical devices and adverse events reporting and 198 reported experiencing an adverse event related to equipment failure. However, 195 nurses were unaware of an official national reporting system for reporting such events. It is important to develop a framework of safe operation of medical devices based on international standards. This reporting system should include the national patients' safety authorities, and should be anonymous, confidential, and non-punitive.

## Introduction

Intensive care units (ICUs) are complex multidisciplinary work environments. Efficient workflow in such areas depends on all team members' knowledge and skills, including the medical and nursing staff, clinical pharmacists, respiratory therapists, dieticians, porters, and technicians, whose integrated efforts result in optimal patient care. This feature of the multidisciplinary team increases patient safety and improves the quality of care in the intensive care unit (ICU).

As intensive care delivery relies heavily on technical resources, the development and use of different devices—including ventilators, vital sign monitors, infusion, and injection pumps, as well as consumables—is fundamental to the integration and functioning of the ICUs. In addition, the safe operation of these devices necessitates the availability of different procedures such

**Funding:** This work was supported by the Deanship of Scientific Research, King Saud University, Riyadh, Saudi Arabia (Research Project No R 17-02-45). The funder had no role in study design, data collection and analysis, decision to publish, or preparation of the manuscript.

**Competing interests:** The authors have declared that no competing interests exist.

as safe use, maintenance, disinfection, reporting of adverse events, and communicating recurrent misuse among users and with manufacturers.[1,2] Locally, there is limited literature on adverse events related to medical devices. In 2007, the Saudi Food and Drug Authority (SFDA) launched a national project for establishing a Medical Devices National Registry (MDNR) to create a database for all medical devices, manufacturers, agents, suppliers, and end-users.[3]

However, the development of the MDNR system remains in its infancy [4] as the national law of "Medical Devices Interim Regulation" was Issued by the SFDA Board of Directors on 27 December 2008, and amended on 27 December 2017. [3] In countries with regulated medical device markets, manufacturers of medical devices, users, and their organizations benefit from reporting adverse events by updating them with latent hazards and hazardous situations of these devices, while they are being used in real life, which allows them to apply the required corrective actions, since risk management is obligatory in these countries for all parties involved in the lifecycle of medical devices. This is an iterative process for the manufacturers, even after placing the device in the market. [5–7]

Therefore, this study aims to assess and explore the level of awareness of ICU nurses in reporting adverse events related to medical devices and how frequently they report adverse events.

## Methods

This is a single center cross-sectional questionnaire-based survey that was conducted in February 2018 and targeted nurses working in the ICUs of King Saud University Medical City (KSUMC) in Riyadh, Saudi Arabia. The required representative sample comprised 218 nurses out of the 502 nurses working in the critical care units (a 95% confidence and a margin of error equal to 5%).

The survey (S1 Form.) was distributed to all the critical care units in the hospital including surgical ICU, Medical ICU, Coronary Care Unit (CCU), High dependency Unit (HDU), Pediatric ICU (PICU) and Neonatal ICU (NICU). The nurses were invited to fill the survey.

The questionnaire was drafted by the authors based on a review of the literature regarding the adverse events related to medical devices. A combination of evidence appraisal together with expert opinion were used in order to draft a suitable questionnaire for this study. MEDLINE® and PubMed searches were performed using the search terms "medical", "device", "reporting", "safety", "usability" and "adverse events OR critical care'". The reference lists of identified papers were screened to identify other relevant papers. After that, four publications were utilized to draft the questionnaire. [8–11] Then a multidisciplinary team focus group meeting was conducted to produce the final version (Appendix I). Experts from biomedical engineering department, Nursing department and pediatric critical care unit reviewed the questionnaire. It was then piloted in our department, and tested to ensure and determine its clarity, before sending it to the targeted group.

### Ethics statement

Participants provided their written informed consent to participate in this study on the first page of the questionnaire (Appendix I). This study was done as quality improvement project at our hospital and received ethical approval by the Institutional Review Board (IRB) of King Saud University. Participants were informed that participation was voluntary and were assured of the confidentiality of their responses and the survey was an anonymous questionnaire. Consent from the participants was obtained.

## The questionnaire

A self-administered questionnaire was designed to explore how the nurses approach different technical issues related to medical devices. The medical devices indicated in the questionnaire were the devices related to the critical care settings, such as mechanical ventilators, defibrillator, cardiopulmonary monitors, and infusion pumps. The questions were grouped into the following three categories:

- The first part of the questionnaire contained questions regarding the participants' demographic data such as: age, gender, credentials, discipline, and length of experience.

- Next, the participants were requested to indicate whether they had ever experienced an adverse event related to the use of a medical device and whether they are aware about any national reporting system for such events. Appropriate response was defined as "examining the device as per the manufacturer's guidelines (User Manual) as received during the user training on a particular device".

- Finally, they were requested to indicate the sequence of steps to manage issues related to any of the devices in the ICU.

## Statistical analysis

Nurses were asked to select the steps they followed when they encountered an adverse event related to the medical devices in the ICU (from the first to the sixth step, starting with the item of priority). The frequencies and percentages of nurses who selected each action were computed and the relative importance index (RII) was used to compute an overall relative order, which we ranked ascendingly to understand the nurses' sequence of action (lower Relative Importance index denotes higher orderly importance). The RII was computed using the method devised by Aziz et al. [12] by summing the individual item responses and dividing them by the product of N × the maximum possible scores.

# Results

## Demographic data

A total of 297 nurses out of 502 nurses working in the critical care units of King Saud University Medical City responded to the survey which makes a response rate of 59%, this rate exceeded the required calculated needed sample size of 218 respondents. The percentage of women was high in the sample (93.3%). Most of the nurses had 6–10 years working experience (26.3%), followed by those with either 3–5 years' or more than 10 years' experience (24.9% in each case), and those who had 1–2 years' experience (23.9%). The majority of nurses were clinical bedside nurses (99%) and only three head nurses responded to the survey. Their working clinical units are shown in Table 1.

## Experiencing adverse events

One hundred ninety eight nurses (66.7%) reported that they had experienced an adverse event related to equipment failure (sudden errors, malfunctions, or shutdown) while using the device. However, 195 nurses (65.7%) were not aware of an official national reporting system for reporting adverse events related to equipment failure. Among the nurses who were aware of the national reporting system, only 44 (14.8%) had reported at least one equipment-related adverse event. Of the 198 who had experienced an equipment-related adverse event, only 44 had reported the incidence, implying that approximately 77.8% of encountered events may go

**Table 1. Respondents' demographic and professional characteristics N = 297.**

|  | Frequency | Percentage |
|---|---|---|
| **Sex** |  |  |
| Female | 277 | 93.3 |
| Male | 20 | 6.7 |
| **Experience in years** |  |  |
| 1–2 years | 71 | 23.9 |
| 3–5 years | 74 | 24.9 |
| 6–10 years | 78 | 26.3 |
| > 10 years | 74 | 24.9 |
| **Clinical Role** |  |  |
| Nurse | 294 | 99 |
| Head Nurse | 3 | 1 |
| **Discipline/working unit** |  |  |
| Medical ICU | 32 | 10.8 |
| Surgical ICU | 48 | 16.2 |
| HDU | 25 | 8.4 |
| CCU | 30 | 10.1 |
| PICU | 87 | 29.3 |
| NICU | 41 | 13.8 |

ICU: Intensive Care Unit, HDU: High Dependency Unit, CCU: Coronary Care Unit, PICU: Pediatric Intensive Care Unit, NICU: Neonatal Intensive Care Unit.

unreported. Additionally, only 31 nurses (10.4%) received feedback from their hospital superiors regarding the reported adverse events. (Table 2) No correlation was found between the length of respondents' experience or age with their knowledge about the existence of that national database (S1 Table.).

## Approach to malfunction

As shown in Table 3, the first action selected by most nurses (89.2%) when they experienced an equipment-related adverse event, in terms of relative importance weight, was to examine

**Table 2. Items in the questionnaire and the frequency and percentage of responses N = 297.**

|  | Frequency | Percentage |
|---|---|---|
| **In the past year, have you encountered a sudden malfunction EVENT with these devices*** |  |  |
| Yes | 198 | 66.7 |
| No | 99 | 33.3 |
| **Are you aware of any official national reporting system regarding these ADVERSE EQUIPMENT events?** |  |  |
| Yes | 102 | 34.3 |
| No | 195 | 65.7 |
| **Have you ever reported an adverse occurrence failure of any equipment,** |  |  |
| Yes | 44 | 14.8 |
| No | 253 | 85.2 |

*****Malfunction was defined as a sudden user error, device malfunction, or sudden shut down of the machine.**

**Table 3. Nurses' actions ranked in the order of a case of sudden equipment failure/adverse event. N = 297.**

|  | 1st step | 2nd step | 3rd step | 4th step | 5th step | 6th step | RII (%) | Order |
|---|---|---|---|---|---|---|---|---|
| Respond to it appropriately by examination | *265 (89.2%)* | 18 (6.1%) | 4 (1.3%) | 4 (1.3%) | 6 (2%) | 4 (1.3) | 20.8 | *1* |
| Contact the superior user or the supervisor | 7 (2.4%) | *161 (54.2%)* | 98 (33%) | 27 (9.1%) | 4 (1.3%) | 0 | 42.1 | *2* |
| Call the biomed technician | 0 | 10 (3.4%) | 120 (40.4%) | *48.8%)* | 19 (6.4%) | 3 (1.6%) | 60.2 | *4* |
| Call the company | 0 | 6 (2%) | 2 (0.7%) | 78 (26.3%) | *184 (62%)* | 27 (9.1%) | 79.2 | *5* |
| Turn it off | 23 (7.7%) | *94 (31.6%)* | 65 (21.9%) | 39 (13.1%) | 69 (23.2%) | 7 (2.7%) | 53.3 | *3* |
| Other actions | 2 (0.7%) | 8 (2.7%) | 12 (4%) | 4 (1.3%) | 15 (5.1%) | *256 (86.2%)* | 94.3 | *6* |

* Lower Relative Importance index denotes higher order of importance.

the device per the guidelines (RII = 20.8%). The next steps selected by the nurses were contacting the superior user/supervisor (RII = 42.1%), turning off the device to restart it (RII = 53.3%), calling the biomedical engineers (RII = 60.2%), and calling the vendor of the device (RII = 79.2%)

## Reporting adverse events

Regarding the reporting of medical equipment failures and adverse events, the Saudi Ministry of Health was selected by 51.1% of the nurses as their first option to report these events, followed by the in-hospital quality improvement and biomedical engineering departments through the hospital built in Electronic Occurrence Variance Report (E-OVR) system (30.8%), the manufacturer via their representative (13.2%), the Saudi Council for Health Specialties (11.0%), the Central Board for Accreditation of Healthcare Institutions (CBAHI) (9.9%), the Saudi Council of Cooperative Health Insurance (3.3%), and the Saudi Food and Drug Authority (2.2%). See Fig 1.

## Communicating risk information updates

The nurses considered the manufacturer as their primary source of information on adverse equipment events (80.5%), followed by their colleagues (35.7%), medical literature (33.7%),

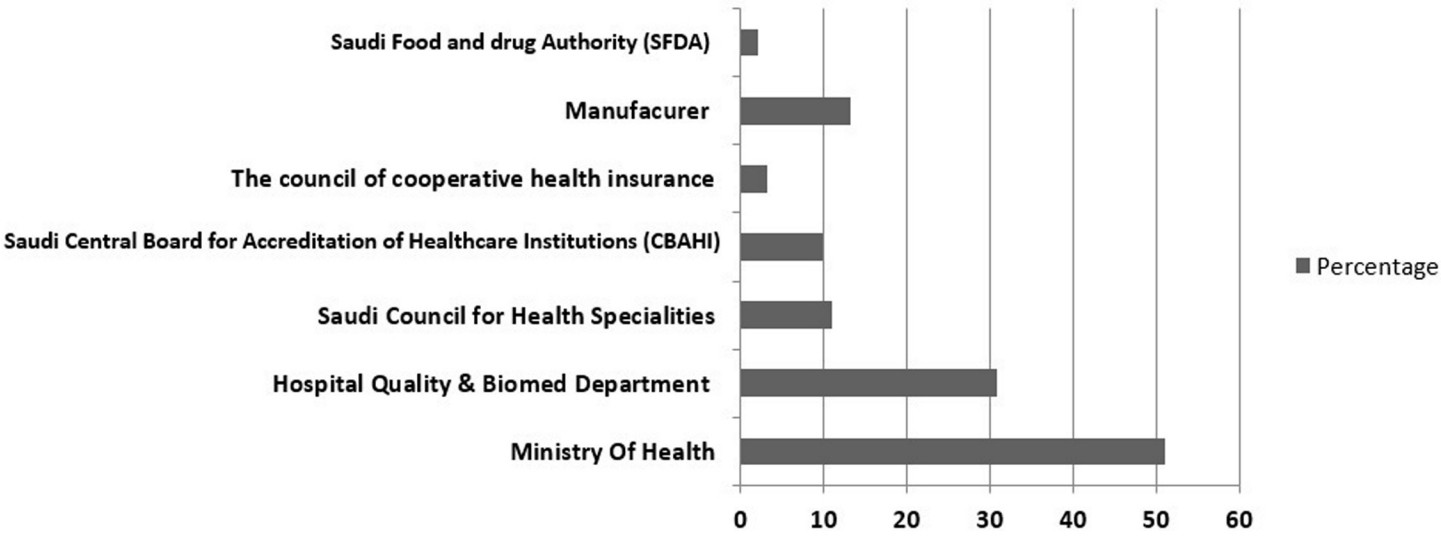

**Fig 1. Nurses reported destinations of reporting equipment related adverse event alerts/incidence.**

and the Saudi Ministry of Health (14.5%). Other less frequently sought sources of information included the hospital quality department adverse occurrence system and biomedical engineering departments' email alerts and memoranda (12.8%), public advertisements (4.5%), and the SFDA (2.0%) (6 respondents out of 279 respondents). See Fig 2.

## Discussion

Reporting of adverse events affecting medical devices is an important factor that contributes massively to their safe operation. The Saudi CBAHI standard only requires this for medical devices in the occurrence of adverse events, while there is neither a clear definition of the meaning of an adverse event nor a clear procedure for reporting these adverse events to a competent national authority. In Saudi Arabia, this consists of a web-based electronic incident reporting system provided by the Medical Devices Sector of the SFDA. [13,14] This could lead to ambiguity since the SFDA defines adverse events related to medical devices as follows:

"*Adverse event: means any malfunction or deterioration in the characteristics and/or performances of a medical device, including any inadequacy in its labeling or the instructions for use that may lead to compromise the health or safety of patients, users or third parties.*"

What is reportable and not reportable is specified through the following definition:

"*Reportable adverse event: means any adverse event or any technical or medical reason leading to a Field Safety Corrective Action, which, directly or indirectly, might lead to or may have led (a) to the death of a patient, a user or another person or (b) to a serious deterioration in their state of health*". [15]

Although over 66% of respondents indicated that they had experienced technical or usage errors related to medical devices, the majority of them (78%) have never reported these adverse events. This low rate could be attributed to the lack of knowledge about the existence of specific legislation for adverse events related to medical devices. [8]

Another possible reason is the fact that healthcare workers might avoid legal and liability consequences that could be traced back to them, because these adverse events might be related to errors in usage and there is no solid protective legislation. [8] This problem has been resolved in different countries with various kinds of legislation by considering anonymous

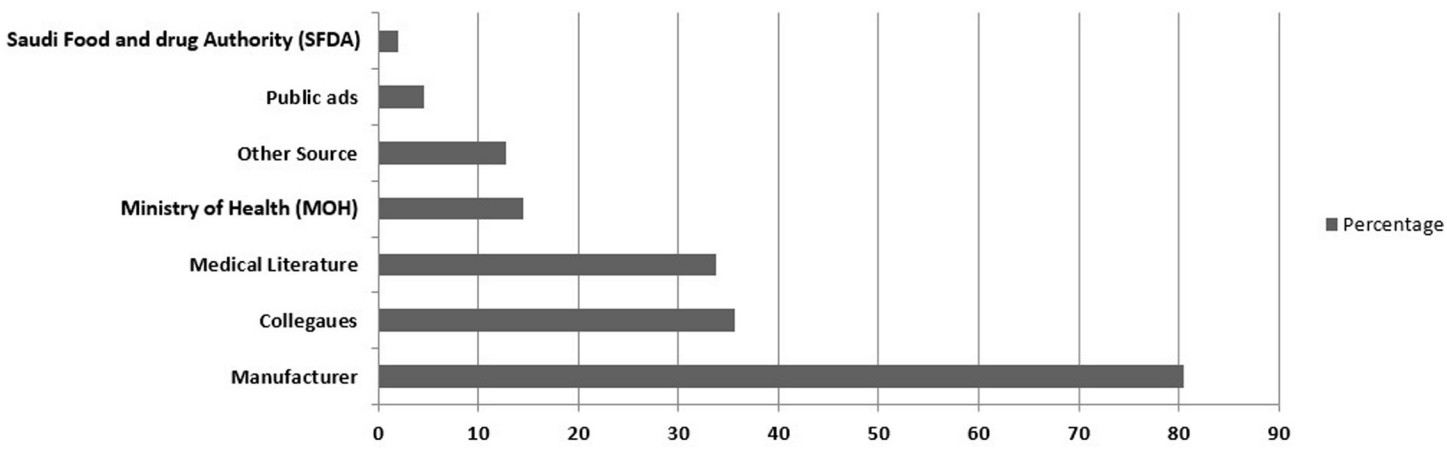

**Fig 2. Nurses Sources of update Information on equipment adverse occurrence alerts.**

reporting and legal protection for the reporter. Anonymous reporting as a feature has been offered either by the MedSun Approach used by the United States Food and Drug Administration (USFDA) or the Danish Health Care System approach. [16,17] Legislation that does not penalize the reporter should be created similar to that of the Danish Act on Patient Safety. [16,18] The MedSun approach is a voluntary medical device adverse event electronic reporting system that guarantees the anonymity of the reporter's personal information by USFDA and maintains contact with the reporter. Each user report will be forwarded in detail to an intermediary appointed by the USFDA MedSun at the user facility, who will review the message report and stay in contact with the reporter to complete the report, to make the report usable from a technical and clinical point of view, to improve the medical device. Another practice to overcome the limitation of anonymous reporting is that of the Danish Health Care System, as they approved and enforced the Act on Patient Safety. This law has been drafted to enhance patient safety and communication of risk information from adverse events by enforcing sanction free reporting. As result of this law, the reporter of adverse events is not subjected to any disciplinary action by his employer or other supervisor or any legal process as a consequence of his/her report. This act was enforced in 2004 and reports showed an increase exceeding 80% within the first two years. [16–18]

In the current context of the national reporting system, it is worth mentioning that errors in usage could result from poor usability research and design by the manufacturer, which could result in recall of the product because usability and human factor issues are considered sources of hazards that need to be eliminated or controlled based on the International Electrotechnical Commission (IEC) requirements. [1, 15, 19–21]

The results show that almost all respondents did not consider the SFDA as a source for medical device risk information. Majority of them relied heavily on the manufacturer feedback. This might be due to a lack of proper communication between the SFDA and the hospital as it has been reported previously in the literature. [8] Another possible cause could be that, although the CBAHI requires healthcare providers to implement a recall management system for medical equipment, they did not mention the SFDA as a source for such information. While for the medication recall management system, the CBAHI incorporated the SFDA in their requirements. [13]

Another striking finding was the low level of awareness of the national reporting system for medical device adverse events among the ICU nurses, as only 2% of respondents reported the correct existing national reporting system, despite that over 66% of the respondents indicated that they have experienced technical or use errors by using medical devices. This might explain why the number of adverse events reported to the SFDA is extremely low compared to the international community. Per a report from the SFDA, the number of adverse event reports received by the SFDA slightly increased from 2 in 2008 to 26 in 2011, then increased to 410 reports in 2012, while in 2013 the number of reports dropped to 68, with a slight increase to 87 in 2016, until it reached 114 in 2017 (SFDA 2013, 2016, 2017). The total number of adverse events reported, per these statistics from 2008 to 2013 and from 2016 to 2017, could not exceed 726 reports. In 2007, the USFDA received over 150000 reports. [22] The Medicine and Healthcare Product Regulatory Agency (MHRA) in the UK received about 10000 incidents reports in 2010. [23] This significant low reporting rate of adverse events to the SFDA must be considered as serious, as these events could have a non-negligible negative impact on patient or user safety.

## Study limitations

We considered the usage error as one of the possible technical problem that might be faced by the healthcare worker as this might result from poor usability research and design by the

manufacturer. The usability and human factor issues are considered sources of hazards that need to be eliminated or controlled based on the International Electrotechnical Commission (IEC) requirements. [1]

While this study highlights significant medical device reporting system from one hospital, this needs further exploration in the other healthcare facilities. The self-reporting nature of this survey may be subjective to recall bias that needs further direct observations in prospective, ICU-based clinical trials.

## Conclusion and recommendations

This study has raised the need to develop a framework for the safe operation of medical devices, based on international standards, that has the following characteristics:

- The Health care workers need to be familiar with the existence of a national reporting system of adverse events to the SFDA. Legislation that does not penalize the reporter should be created.

- The role of the Medical Devices SFDA-Officer for healthcare providers should be activated (Saudi Food & Drug Authority 2018).

- The internal E-OVR in the hospitals should be directed to the Saudi Center for Patient Safety and the center will direct the report to the SFDA.

- Recalls for medical devices should be directed from the SFDA through the Saudi Center for Patient Safety and disseminated to hospitals.

- A database should be created where risk information is categorized and made accessible with multiple interfaces for:

  • Medical device users (Voluntary)–the aim of this database is to facilitate networking and sharing experiences among users

  • Medical device manufacturers or designers (mandatory)

  • Healthcare organizations (such as for risk management departments) (mandatory).

## Supporting information

**S1 Form. This is the S1 Form of the questionnaire of the survey.**
(PDF)

**S1 Table. This is the S1 Table where it shows the correlation of the respondents' characteristics and their awareness of the national reporting system for adverse events.**
(XLSX)

## Acknowledgments

We thank all the ICU nurses working at King Saud University Medical City who participated in this study.

## Author Contributions

**Conceptualization:** Fahad Alsohime, Mohamad-Hani Temsah, Ayman Al-Eyadhy, Sanaa Gulman, Haytam Issa.

**Data curation:** Sanaa Gulman, Haytam Issa.

**Formal analysis:** Fahad Alsohime.

**Investigation:** Fahad Alsohime, Haytam Issa.

**Methodology:** Fahad Alsohime, Mohamad-Hani Temsah, Ayman Al-Eyadhy.

**Resources:** Omar Alsohime.

**Software:** Omar Alsohime.

**Supervision:** Fahad Alsohime, Mohamad-Hani Temsah, Ayman Al-Eyadhy.

**Visualization:** Omar Alsohime.

**Writing – original draft:** Fahad Alsohime, Mohamad-Hani Temsah, Gamal Hasan, Ayman Al-Eyadhy.

**Writing – review & editing:** Fahad Alsohime, Mohamad-Hani Temsah, Gamal Hasan, Ayman Al-Eyadhy.

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
