## [Decision Letter · Decision Letter 0]

7 Aug 2019

PONE-D-19-19700

Reporting adverse events related to medical devices: A single center experience from a tertiary academic hospital

PLOS ONE

Dear Dr Alsohime,

Thank you for submitting your manuscript to PLOS ONE. After careful consideration, we feel that it has merit but does not fully meet PLOS ONE’s publication criteria as it currently stands. Therefore, we invite you to submit a revised version of the manuscript that addresses the points raised during the review process.

We would appreciate receiving your revised manuscript by Sep 21 2019 11:59PM. To enhance the reproducibility of your results, we recommend that if applicable you deposit your laboratory protocols in protocols.io, where a protocol can be assigned its own identifier (DOI) such that it can be cited independently in the future. For instructions see: http://journals.plos.org/plosone/s/submission-guidelines#loc-laboratory-protocols

We look forward to receiving your revised manuscript.

Kind regards,

Lars-Peter Kamolz, M.D., Ph.D., M.Sc.

Academic Editor

PLOS ONE

Journal Requirements:

2. Please include additional information regarding the survey or questionnaire used in the study and ensure that you have provided sufficient details that others could replicate the analyses. For instance, if you developed a questionnaire as part of this study and it is not under a copyright more restrictive than CC-BY, please include a copy, in both the original language and English, as Supporting Information.  If the original language is written in non-Latin characters, for example Amharic, Chinese, or Korean, please use a file format that ensures these characters are visible.

3. Please amend your current ethics statement to address the following concerns:  

a) Did participants provide their written or verbal informed consent to participate in this study?

Reviewers' comments:

Reviewer's Responses to Questions

**Comments to the Author**

1. Is the manuscript technically sound, and do the data support the conclusions?

Reviewer #1: Partly

Reviewer #2: Yes

2. Has the statistical analysis been performed appropriately and rigorously? 

Reviewer #1: I Don't Know

Reviewer #2: Yes

3. Have the authors made all data underlying the findings in their manuscript fully available?

Reviewer #1: No

Reviewer #2: Yes

4. Is the manuscript presented in an intelligible fashion and written in standard English?

Reviewer #1: Yes

Reviewer #2: Yes

5. Review Comments to the Author

Reviewer #1: Reviewer’s comments

A reporting system was described in this manuscript. However, there are some issues that need to be addressed.

Major comments

1. There is little description about the national reporting system, the name of the law, the time of legislation, the details in the required reporting, etc.

2. Demographic characteristics: the nationality/citizenship was not reported. Howe many of them were native? Do they have difficulties in understanding the regulations?

3. Can the authors be more specific about the “devices”? What are the devices that are more frequently used in the settings?

4. How was “Respond to it appropriately by examination” in the questionnaire defined?

5. The reporting interface (?) or methods were not described. In the Results, it seemed complicated. Was the reporting part of nursing education?

6. In the Discussion, the explanations about the key terms should be moved to the Introduction or the Methods.

Minor comments

1. In the first sentence, Line 67, what does “(a 95% confidence interval and 5% marginal error)” mean?

2. In the Discussion, was there any data about the reporting rates in previous studies?

Reviewer #2: Dear authors,

I got to review your manuscript "Reporting adverse events related to medical devices: A single center experience from a tertiary academic hospital". In my opinion, your manuscript is already in a quite good state dealing with a topic highly worthwhile to further assess. However, there are some aspects, that should be covered/questions that were raised during the reading:

1.) In your introduction, line 53, you state that the development of the MDNR system remains in its infancy. However, you do not further elaborate. Please write one or two sentences as to why this system is not that well developed yet.

2.) In line 71 you state, that the questionnaire was drafted by the authors based on a review of the literature - can you further elaborate as to how this review was assessed? Do you refer to specific papers?

3.) Please keep the same order in Methods (1, 2, 3) and in results/discussion (1, 3, 2). I recommend to first present the results of the step sequence and THEN the personal experience (as it is in methods section).

4.) Have there been any nurse pupils, assistance care givers been questioned, as well?

5.) Please delete "approximately" in line 118, as it makes no sense.

6.) I recommend to give the percentage of those who had reported an AE compared to those who were aware of the system, as you state in the sentence (line 121f.). Then you can also lose the next sentence.

7.) In table 2, why does it say n=103 at the third question? Does that not have to be 102? And further, if n=102, how come the frequency is higher? I would only indicate the nurses, that have ticked yes in the second question (therefore n=102; yes: 44, No: 58).

8.) In your sequence of steps, I do miss the relevance of the patient - how comes, that there is no step "securing the patient's survival" or similar?

9.) In table 3 the n at "call the biomed technician" is missing.

10.) On page 12 in your discussion, you state, that the nurses had also reported usage errors - however, prior to that there is no mentioning of usage errors, just technical errors. So I recommend to delete it here AND

11.) Introduce a limitations section, in which you could also discuss, that some of these technical errors could be due to wrong usage and so on. In general, a limitations section would be helpful.

12.) In the middle of page 14, where do the 2% come from? I thought it would have been n=102, therefore around 34%? If the rest could not name the right system, then please introduce these 2% and this fact in results section already.

13.) Have you found any correlation between length of experience/age with the knowledge of the existence of that database? Please add this to your manuscript.

14.) Have you found any literature concerning the same topic with other medical professions? E.g. doctors - I would be interested in how many of those know about the existence of said systems. Please discuss.

Thank you.

6. PLOS authors have the option to publish the peer review history of their article (what does this mean?). If published, this will include your full peer review and any attached files.

Reviewer #1: No

Reviewer #2: No

---

## [Author Response · Author response to Decision Letter 0]

28 Aug 2019

Dear Dr. Kamolz

We would like to thank you and the reviewers for the comments and suggestions pertaining to the manuscript #PONE-D-19-19700: “Reporting adverse events related to medical devices: A single center experience from a tertiary academic hospital” in PLOS ONE

We have now addressed all points, as detailed below.

Reviewer #1: Reviewer’s comments

A reporting system was described in this manuscript. However, there are some issues that need to be addressed.

Major comments

1. There is little description about the national reporting system, the name of the law, the time of legislation, the details in the required reporting, etc.

Thank you for your comment. The name of the national law is the “Medical Devices Interim Regulation“. This law was issued by the Saudi Food and Drug Authority (SFDA) Board of Directors decree number (1-8-1429) and dated 27 December 2008, and amended by SFDA Board of Directors decree number (4-16-1439) dated 27 December 2017 

(https://www.sfda.gov.sa/ar/medicaldevices/regulations/Documents/MD-InterimRegulation-en.pdf)

The details in the required reporting can be found in the folowing link

(https://ncmdr.sfda.gov.sa/Default.aspx)

Changes done in the text.

2. Demographic characteristics: the nationality/citizenship was not reported. Howe many of them were native? Do they have difficulties in understanding the regulations?

Thank you for raising this point. We did not enquire about the nationality to avoid creating bias among the healthcare workers. All of them speak English fluently as first or second language. The questionnaire was constructed in English knowing that the official language of the healthcare system in Saudi Arabia is the English language and all the legislative regulations related to the healthcare are issued in both English and Arabic.

3. Can the authors be more specific about the “devices”? What are the devices that are more frequently used in the settings?

Thank you for this comment. The medical devices indicated in the questionnaire were the devices related to the critical care settings; such as mechanical ventilators, defibrillator, cardiopulmonary monitors, infusion pumps,.. etc. 

Changes done in the text.

4. How was “Respond to it appropriately by examination” in the questionnaire defined?

Thank you for your comment. Appropriate response was defined as “examining the device as per the manufacturer‘s guidelines (User Manual) as received during the user training on a particular device“.

Changes done in the text.

5. The reporting interface (?) or methods were not described. In the Results, it seemed complicated. Was the reporting part of nursing education?

Thank you for your point. Indeed, the reporting interface is an electronic website (https://ncmdr.sfda.gov.sa/Default.aspx) designed for healthcare workers to report the adverse events. Unfortunately, it is not included in the nursing education and this study aimed to support the need to integrate this reporting system the healthcare education.

6. In the Discussion, the explanations about the key terms should be moved to the Introduction or the Methods.

Thank for your comment. In the beginning of the discussion, we wanted to elaborate more on the adverse events related to medical devices as this would help the reader to better understand the evolution of this system over the last several years. 

Minor comments

1. In the first sentence, Line 67, what does “(a 95% confidence interval and 5% marginal error)” mean?

Sorry for the typo, it was corrected to:

“95% confidence and a margin of error equal to 5%”

The 95% level is the most commonly used. If the level of confidence is 95%, the "true" percentage for the entire population would be within the margin of error around a poll's reported percentage 95% of the time.

http://theknowledge.site/wp/m/Margin_of_error.htm

2. In the Discussion, was there any data about the reporting rates in previous studies?

Unfortunately, studies in this field are scarce and extremely limited. The existing source of the reporting rate of adverse events related to medical device over the last several years is the annual statistical report released by the Saudi Food and Drug Authority SFDA

Reviewer #2: Dear authors,

I got to review your manuscript "Reporting adverse events related to medical devices: A single center experience from a tertiary academic hospital". In my opinion, your manuscript is already in a quite good state dealing with a topic highly worthwhile to further assess. However, there are some aspects, that should be covered/questions that were raised during the reading:

1.) In your introduction, line 53, you state that the development of the MDNR system remains in its infancy. 

However, you do not further elaborate. Please write one or two sentences as to why this system is not that well developed yet.

Thank you for your comment. This was elaborated now in the text, mentionioning that the national law “ Medical Devices Interim Regulation“ was Issued by the Saudi Food and Drug Authority Board of Directors decree number (1-8-1429) and dated 27 December 2008, and amended by Saudi Food and Drug Authority Board of Directors decree number (4-16-1439) dated 27 December 2017 

(https://www.sfda.gov.sa/ar/medicaldevices/regulations/Documents/MD-InterimRegulation-en.pdf)

2.) In line 71 you state, that the questionnaire was drafted by the authors based on a review of the literature - can you further elaborate as to how this review was assessed? Do you refer to specific papers?

A combination of evidence appraisal together with expert opinion were used in order to draft a suitable questionnaire for this study.

MEDLINE® and PubMed searches were performed using the search terms “medical”, “device”, “reporting”, “safety” “ Usability” and “‘adverse events OR critical care’”. The reference lists of identified papers were screened to identify other relevant papers.

*Teow N and Siegel SJ. FDA Regulation of Medical Devices and Medical Device Reporting. Pharmaceut Reg Affairs. 2013;2:110.

* Geissler N, Byrnes T, Lauer W, Radermacher K, Kotzsch S, Korb W, Hölscher UM. Patient safety related to the use of medical devices: a review and investigation of the current status in the medical device industry. Biomed Tech (Berl). 2013;58(1):67-78.

* Mattox E. Medical devices and patient safety. Crit Care Nurse. 2012;32(4):60-8.

* Rich S. How human factors lead to medical device adverse events. Nursing. 2008 Jun;38(6):62-3.

3.) Please keep the same order in Methods (1, 2, 3) and in results/discussion (1, 3, 2). I recommend to first present the results of the step sequence and THEN the personal experience (as it is in methods section).

Thank for your comment. In the beginning of the discussion, we wanted to elaborate more on the adverse events related to medical devices as this would help the reader to better understand the evolution of this system over the last several years. 

4.) Have there been any nurse pupils, assistance care givers been questioned, as well?

No, in our study we included only the practicing nurses.

5.) Please delete "approximately" in line 118, as it makes no sense.

Thanks for highlighting this, it was deleted in the revised manuscript. 

6.) I recommend to give the percentage of those who had reported an AE compared to those who were aware of the system, as you state in the sentence (line 121f.). Then you can also lose the next sentence.

The aim of the study is to explore the awareness of the correct national reporting system, that’s why the question about if they ever reported an adverse event or not was limited only to the 102 nurses who reported that they were aware of the national reporting system as we assumed that those nurses who were not aware about the reporting system were never exposed to an event that lead to report this event. So only 44 nurses (14.8%) among the 102 nurses who were aware of the reporting system had reported at least one equipment-related adverse event over the last year.

7.) In table 2, why does it say n=103 at the third question? Does that not have to be 102? And further, if n=102, how come the frequency is higher? I would only indicate the nurses, that have ticked yes in the second question (therefore n=102; yes: 44, No: 58).

Thank you. Indeed, this is a typing error. Correction carried out

8.) In your sequence of steps, I do miss the relevance of the patient - how comes, that there is no step "securing the patient's survival" or similar?

Thank you for this important point. The patient safety is always included in the Appropriate response to a device malfunction as per the manufacturer guidelines (User Manual). Changes done in the text.

9.) In table 3 the n at "call the biomed technician" is missing.

The typo was corrected.

10.) On page 12 in your discussion, you state, that the nurses had also reported usage errors - however, prior to that there is no mentioning of usage errors, just technical errors. So I recommend to delete it here AND

11.) Introduce a limitations section, in which you could also discuss, that some of these technical errors could be due to wrong usage and so on. In general, a limitations section would be helpful.

Thank you. We considered the usage error as one of the possible technical error that might be faced by the healthcare worker and this was indicated in the questionnaire. Correction carried out in the text.

We considered the usage error among the technical problems as poor usability research and design by the manufacturer, which could result in recall of the product because usability and human factor issues are considered sources of hazards that need to be eliminated or controlled based on the International Electrotechnical Commission (IEC) requirements

12.) In the middle of page 14, where do the 2% come from? I thought it would have been n=102, therefore around 34%? If the rest could not name the right system, then please introduce these 2% and this fact in results section already.

6 respondents out of 279 respondents considered the Saudi FDA the source of risk information such as recalls.

13.) Have you found any correlation between length of experience/age with the knowledge of the existence of that database? Please add this to your manuscript.

No correlation has been found. This information has been added to the text

14.) Have you found any literature concerning the same topic with other medical professions? E.g. doctors - I would be interested in how many of those know about the existence of said systems. Please discuss.

Unfortunately, studies in this field are scarce and extremely limited. 

We included the nurses in the critical areas as they are the frontliners who are dealing first with these technical issues

---

## [Decision Letter · Decision Letter 1]

20 Sep 2019

PONE-D-19-19700R1

Reporting adverse events related to medical devices: A single center experience from a tertiary academic hospital

PLOS ONE

Dear Dr Alsohime,

Thank you for submitting your manuscript to PLOS ONE. After careful consideration, we feel that it has merit but does not fully meet PLOS ONE’s publication criteria as it currently stands. Therefore, we invite you to submit a revised version of the manuscript that addresses the points raised during the review process.

We would appreciate receiving your revised manuscript by Nov 04 2019 11:59PM. To enhance the reproducibility of your results, we recommend that if applicable you deposit your laboratory protocols in protocols.io, where a protocol can be assigned its own identifier (DOI) such that it can be cited independently in the future. For instructions see: http://journals.plos.org/plosone/s/submission-guidelines#loc-laboratory-protocols

We look forward to receiving your revised manuscript.

Kind regards,

Lars-Peter Kamolz, M.D., Ph.D., M.Sc.

Academic Editor

PLOS ONE

Reviewers' comments:

Reviewer's Responses to Questions

**Comments to the Author**

1. If the authors have adequately addressed your comments raised in a previous round of review and you feel that this manuscript is now acceptable for publication, you may indicate that here to bypass the “Comments to the Author” section, enter your conflict of interest statement in the “Confidential to Editor” section, and submit your "Accept" recommendation.

Reviewer #1: All comments have been addressed

Reviewer #2: (No Response)

2. Is the manuscript technically sound, and do the data support the conclusions?

Reviewer #1: Yes

Reviewer #2: (No Response)

3. Has the statistical analysis been performed appropriately and rigorously? 

Reviewer #1: I Don't Know

Reviewer #2: (No Response)

4. Have the authors made all data underlying the findings in their manuscript fully available?

Reviewer #1: Yes

Reviewer #2: (No Response)

5. Is the manuscript presented in an intelligible fashion and written in standard English?

Reviewer #1: Yes

Reviewer #2: (No Response)

6. Review Comments to the Author

Reviewer #1: Thank you for the revision.

Major comments

- I am not familiar with the type of statistics used in the manuscript. The fixed effects of the types of ICUs may need to be controlled or specified by the ICUs, because the types of medical devices most frequently used and the levels of training might vary accordingly.

- Discussion: there are several recommendations. However, they are not all based on the findings. For example, “The reporting system of adverse events to the SFDA should be anonymous, confidential, and non-punitive” was unrelated to the findings and inferred based on the comparisons with other jurisdictions. The non-punitive principle was mentioned in the first and second points. The term, “activated”, was not clear.

Minor comments

- Results: please use the tense consistently. Both present and past tenses were used.

- Abstract: “international standard”? Do you mean the Danish system?

- Table 1 and 2: please add total.

- Fig 1 & 2: the units? Abbreviations?

Reviewer #2: Dear authors,

thank you for your revision. In my opinion, there is still some space for improvement. I would like to ask you to consider the following:

1.) While I can imagine, that the development of the MDNR system remains in its infancy because it has just been amended, it still remains elusive to me, if you mean that. If so, please indicate (see previous comment 1.)

2.) Even if you want to elaborate more on the adverse events in the discussion, the logical flow is compromised if you switch the orders. If you want to discuss the personal experience before the approach, please switch the order in the methods section then. (see previous comment 3)

3.)In table 2, the third question still makes no sense to me. How can the frequency of no be 253, if the whole n is just 102? Again a typo? Should it be 58? (See previous comment 7).

4.) Even if you explained the 2% (6 out of 279) now, you should really introduce that number in the results already. I do not want to think about the numbers again, when I am in discussion already. Every "new" result just leads to confusion. (see previous comment 12).

5.) You should really introduce a clearly delineated limitations section. (see previous comment 11)

Thank you.

7. PLOS authors have the option to publish the peer review history of their article (what does this mean?). If published, this will include your full peer review and any attached files.

Reviewer #1: No

Reviewer #2: No

---

## [Author Response · Author response to Decision Letter 1]

23 Sep 2019

Dear Dr. Kamolz

We would like to thank you and the reviewers for the comments and suggestions pertaining to the manuscript #PONE-D-19-19700: “Reporting adverse events related to medical devices: A single center experience from a tertiary academic hospital” in PLOS ONE

We have now addressed all points, as detailed below.

Reviewer #1: Thank you for the revision.

Major comments

1- I am not familiar with the type of statistics used in the manuscript. The fixed effects of the types of ICUs may need to be controlled or specified by the ICUs, because the types of medical devices most frequently used and the levels of training might vary accordingly.

Thanks dear reviewer for the above comment. Regarding the first part, we suggest to refer to the below studies which can explain the type of the statics that we applied for our study.

For the second part in term of fixed effects of the types of the ICUs, we conducted a descriptive, qualitative and quantitative analysis for universal parameters and concepts applicable for the different types of the ICUs. Investigated in this study. 

Aziz, N., Zain, Z., Mafuzi, R. M. Z. R., Mustapa, A. M., Najib, N. H. M., & Lah, N. F. N. (2016). Relative importance index (RII) in ranking of procrastination factors among university students.

Holt, G. D. (2014). Asking questions, analysing answers: relative importance revisited. Construction Innovation, 14(1), 2–16.

2- Discussion: there are several recommendations. However, they are not all based on the findings. For example, “The reporting system of adverse events to the SFDA should be anonymous, confidential, and non-punitive” was unrelated to the findings and inferred based on the comparisons with other jurisdictions. The non-punitive principle was mentioned in the first and second points. The term, “activated”, was not clear.

Thank you for your observation, adjustments carried out as it appears in the below text and these changes were done in the manuscript.

Many thanks 

Conclusion and Recommendations 

This study has raised the need to develop a framework for the safe operation of medical devices, based on international standards, that has the following characteristics:

- The Health care workers need to be familiar with the existence of a national reporting system of adverse events to the SFDA. Legislation that does not penalize the reporter should be created.

- The role of the Medical Devices SFDA-Officer for healthcare providers should be activated (Saudi Food & Drug Authority 2018). 

- The internal E-OVR in the hospitals should be directed to the SFDA. 

- Recalls for medical devices should be directed from the SFDA through the Saudi Center for Patient Safety and disseminated to hospitals. 

- A database should be created where risk information is categorized and made accessible with multiple interfaces for:

• Medical device users (Voluntary) – the aim of this database is to facilitate networking and sharing experiences among users

• Medical device manufacturers or designers (mandatory)

• Healthcare organizations (such as for risk management departments) (mandatory).

Minor comments

1- Results: please use the tense consistently. Both present and past tenses were used.

Thank you for your remark, changes done accordingly in the text.

2- Abstract: “international standard”? Do you mean the Danish system?

Mainly we refer to the standards implemented in the western countries such as the United States Food and Drug Administration (USFDA) or the Danish Health Care System approach.16-17 

3- Table 1 and 2: please add total.

Indeed, we added the total 

4- Fig 1 & 2: the units? Abbreviations?

Thank you for your kind observation. Changes carried out. 

Reviewer #2: 

thank you for your revision. In my opinion, there is still some space for improvement. I would like to ask you to consider the following:

1.) While I can imagine, that the development of the MDNR system remains in its infancy because it has just been amended, it still remains elusive to me, if you mean that. If so, please indicate (see previous comment 1.)

Thank you for your notion. Exactly, as the national law of “ Medical Devices Interim Regulation” was Issued by the SFDA Board of Directors on 27 December 2008, and amended on 27 December 2017 (change in the text carried out)

And this was elaborated from Ref (No 4) too that states in their abstract: “However, the progress of medical device registry systems and post-market medical device surveillance systems remains in its infancy in Saudi Arabia and within the region. In 2007, a royal decree assigned the responsibility for regulating medical devices to the Saudi Food and Drug Authority (SFDA). Soon afterwards, the SFDA established the Medical Devices National Registry (MDNR) to house medical device information relating to manufacturers, agents, suppliers and end-users. 

2.) Even if you want to elaborate more on the adverse events in the discussion, the logical flow is compromised if you switch the orders. If you want to discuss the personal experience before the approach, please switch the order in the methods section then. (see previous comment 3)

 Thank you for this remark. We totally agree with you and we switched the order in the method section. 

3.)In table 2, the third question still makes no sense to me. How can the frequency of no be 253, if the whole n is just 102? Again a typo? Should it be 58? (See previous comment 7).

Our sincere apology, that was a typing error. 

The question “Have you ever reported an adverse occurrence failure of any equipment?” Was directed to all the nurses, 44 of them indicated ‘yes’, and 253 answered ‘No’ 

4.) Even if you explained the 2% (6 out of 279) now, you should really introduce that number in the results already. I do not want to think about the numbers again, when I am in discussion already. Every "new" result just leads to confusion. (see previous comment 12).

Indeed, This was moved to the Results section, thank you for the advice. 

5.) You should really introduce a clearly delineated limitations section. (see previous comment 11)

This is an excellent advice. It was added to the text

 Study limitations:

We considered the usage error as one of the possible technical problem that might be faced by the healthcare worker as this might result from poor usability research and design by the manufacturer. The usability and human factor issues are considered sources of hazards that need to be eliminated or controlled based on the International Electrotechnical Commission (IEC) requirements. 1

While this study highlights significant medical device reporting system from one hospital, this needs further exploration in the other healthcare facilities. The self-reporting nature of this survey may be subjective to recall bias that needs further direct observations in prospective, ICU-based clinical trials.

---

## [Decision Letter · Decision Letter 2]

9 Oct 2019

Reporting adverse events related to medical devices: A single center experience from a tertiary academic hospital

PONE-D-19-19700R2

Dear Dr. Alsohime,

We are pleased to inform you that your manuscript has been judged scientifically suitable for publication and will be formally accepted for publication once it complies with all outstanding technical requirements.

With kind regards,

Lars-Peter Kamolz, M.D., Ph.D., M.Sc.

Academic Editor

PLOS ONE

Additional Editor Comments (optional):

Reviewers' comments:

Reviewer's Responses to Questions

**Comments to the Author**

1. If the authors have adequately addressed your comments raised in a previous round of review and you feel that this manuscript is now acceptable for publication, you may indicate that here to bypass the “Comments to the Author” section, enter your conflict of interest statement in the “Confidential to Editor” section, and submit your "Accept" recommendation.

Reviewer #1: All comments have been addressed

Reviewer #2: All comments have been addressed

2. Is the manuscript technically sound, and do the data support the conclusions?

Reviewer #1: Yes

Reviewer #2: (No Response)

3. Has the statistical analysis been performed appropriately and rigorously? 

Reviewer #1: Yes

Reviewer #2: (No Response)

4. Have the authors made all data underlying the findings in their manuscript fully available?

Reviewer #1: Yes

Reviewer #2: (No Response)

5. Is the manuscript presented in an intelligible fashion and written in standard English?

Reviewer #1: Yes

Reviewer #2: (No Response)

6. Review Comments to the Author

Reviewer #1: Thank you for the revision.

Please revise the following items:

Typos to remove:

1. Line 112: remove 4.

Reviewer #2: (No Response)

7. PLOS authors have the option to publish the peer review history of their article (what does this mean?). If published, this will include your full peer review and any attached files.

Reviewer #1: No

Reviewer #2: No

---

## [Editor Report · Acceptance letter]

16 Oct 2019

PONE-D-19-19700R2 

Reporting adverse events related to medical devices: A single center experience from a tertiary academic hospital 

Dear Dr. Alsohime:

I am pleased to inform you that your manuscript has been deemed suitable for publication in PLOS ONE. Congratulations! Your manuscript is now with our production department. 

With kind regards,

on behalf of

Dr. Lars-Peter Kamolz 

Academic Editor

PLOS ONE